# Deterministic Langevin Monte Carlo with Normalizing Flows for Bayesian Inference

**Richard D.P. Grumitt**
Department of Astronomy, Tsinghua University,
Beijing 100084, China

**Biwei Dai**
Physics Department, University of California
Berkeley, CA 94720, USA

**Uroš Seljak**
Physics Department, University of California
and Lawrence Berkeley National Laboratory
Berkeley, CA 94720, USA

## Abstract

We propose a general purpose Bayesian inference algorithm for expensive likelihoods, replacing the stochastic term in the Langevin equation with a deterministic density gradient term. The particle density is evaluated from the current particle positions using a Normalizing Flow (NF), which is differentiable and has good generalization properties in high dimensions. We take advantage of NF preconditioning and NF based Metropolis-Hastings updates for a faster convergence. We show on various examples that the method is competitive against state of the art sampling methods.

## 1 Introduction

The task of Bayesian inference is to determine the posterior $p(\boldsymbol{x}|\boldsymbol{y})$ of $d$ parameters $\boldsymbol{x}$ given the likelihood $p(\boldsymbol{y}|\boldsymbol{x})$ of some data $\boldsymbol{y}$ and given the prior $p(\boldsymbol{x})$, such that the posterior is given by Bayes theorem as $p(\boldsymbol{x}|\boldsymbol{y}) = p(\boldsymbol{y}|\boldsymbol{x})p(\boldsymbol{x})/p(\boldsymbol{y})$. While we have access to the joint $p(\boldsymbol{x}, \boldsymbol{y}) = p(\boldsymbol{y}|\boldsymbol{x})p(\boldsymbol{x})$, the normalization $p(\boldsymbol{y})$ is generally unknown. Bayesian posterior inference can be related to the general task of finding a stationary distribution in an external potential by defining $U(\boldsymbol{x}) = -\ln p(\boldsymbol{y}|\boldsymbol{x}) - \ln p(\boldsymbol{x}) = \mathcal{L}(\boldsymbol{x}) + \mathcal{P}(\boldsymbol{x})$, where $\mathcal{L}(\boldsymbol{x})$ is the negative log likelihood and $\mathcal{P}(\boldsymbol{x})$ the negative log prior. A common approach to this task is to use a particle based Langevin equation, which solves a stochastic differential equation (SDE) for the particle evolution (Langevin Monte Carlo, LMC). A related method is Hamiltonian Monte Carlo (HMC) [Duane et al., 1987], which uses particle positions and velocities to evolve the particles. In these cases the long time equilibrium solution is that of the target distribution, the posterior $p(\boldsymbol{x}|\boldsymbol{y})$. LMC and HMC are two of the most popular gradient based implementations of Markov Chain Monte Carlo (MCMC), but for finite step size they require Metropolis-Hastings (MH) adjustment to become unbiased. In this case they have theoretical convergence guarantees to the stationary distribution (under mild ergodicity assumptions), but suffer from chain element correlations, which may render the convergence to be very slow.

In this paper we propose a Deterministic Langevin Monte Carlo (DLMC) with Normalizing Flows (NF). It uses a random initialization from the prior, but the subsequent evolution is deterministic, replacing the stochastic velocity term in the Langevin equation with a deterministic density gradient term. To avoid the need to solve the Fokker-Planck equation directly we evaluate the density defined by the current particle positions via an NF, and then use its gradient to update the first order particle Langevin dynamics. NFs are differentiable and have good generalization properties in high dimensions.

36th Conference on Neural Information Processing Systems (NeurIPS 2022).

Deterministic sampling methods have until now been limited to particle based methods. Two examples are Stein Variational Gradient Descent (SVGD) [Liu and Wang, 2016], and interacting particle solutions of Maoutsa et al. [2020], which proposes particle based scores for the gradient of the log-density. However, these particle based methods scale poorly to high dimensions. The main novelty of our contribution is the introduction of NFs to the class of deterministic methods for evaluating stationary distributions: we use NFs for density estimation and gradient evaluation, as well as for MH adjustment to correct for imperfect NF density estimation and accelerate convergence.

Our primary target application is to expensive likelihoods (wall clock time of seconds or more). An example are inverse problems in scientific applications, where often one must solve an expensive Ordinary or Partial Differential Equation (ODE or PDE) forward model, where the computational cost can be minutes or hours. In such applications the computational cost of the NF (which is of order seconds in our experiments) is not dominant. For fast model evaluations standard MCMC methods suffice and our target are problems where the cost of standard methods is prohibitive. The computational cost of a forward model is unrelated to the complexity of the posterior distribution: often the posteriors are simple Gaussian distributions even when the computational cost is high. In such settings the main goal of posterior analysis is to minimize the number of likelihood calls to reach some prescribed precision on the posterior.

DLMC has the following advantages in comparison to several other MCMC samplers:

- DLMC takes advantage of gradient information of the target distribution, but without stochastic noise, which slows down the particle evolution towards the target in LMC and HMC. As a result, DLMC gradient based optimization can rapidly move the particles into the region of high posterior mass for a fast convergence.

- DLMC is not a sequential Markov Chain, and DLMC particles can be updated in parallel at each time step, which can take advantage of trivial machine parallelization of likelihood evaluations, reducing the wall-clock time compared to standard MCMC.

- DLMC particles are initialized as a random draw from the prior, which covers the posterior. Each particle produces an independent sample in the limit of a large number of particles ($N$).

- DLMC uses NF for density estimation, and takes advantage of NF preconditioning and NF based MH updates, for a faster convergence and to help correct for imperfect density estimation.

- DLMC can handle multimodal posteriors, in contrast to standard MCMC samplers.

## 2 Langevin and Fokker-Planck equations

The overdamped Langevin equation is a stochastic differential equation describing particle motion in an external potential and subject to a random force with zero mean,

$$\dot{\boldsymbol{x}} = \boldsymbol{v} = -\boldsymbol{\nabla} U(\boldsymbol{x}) + \boldsymbol{\eta}, \tag{1}$$

where $\langle \boldsymbol{\eta}(\boldsymbol{t}) \rangle = 0$ and $\langle \eta_i(t) \eta_j(t') \rangle = 2\delta_{ij}\delta(t - t')$. For simplicity we set the diffusion coefficient, temperature and mobility to unity. The Langevin equation is a first order velocity equation which has a deterministic velocity $-\boldsymbol{\nabla} U(\boldsymbol{x})$ and a stochastic velocity $\boldsymbol{\eta}$. Here the gradient operator is with respect to the parameters of $U(\boldsymbol{x})$, $\boldsymbol{\nabla} U(\boldsymbol{x}) = dU(\boldsymbol{x})/d\boldsymbol{x}$.

In practice we need to discretize the Langevin equation, and for any finite step size $\Delta t$ the result is a biased distribution. An example is the Ornstein-Uhlenbeck process, where $U(\boldsymbol{x})$ is a harmonic potential, $\exp(-U(\boldsymbol{x})) = N(\boldsymbol{x}; \boldsymbol{\mu}, \boldsymbol{\Sigma})$. Standard Langevin algorithm updates lead to the solution $q(\boldsymbol{x}) = N(\boldsymbol{\mu}, \boldsymbol{\Sigma}(I - \Delta t \boldsymbol{\Sigma}^{-1}/2)^{-1})$ [Wibisono, 2018]. We can see that for a finite fixed stepsize $\Delta t$ the solution converges to a biased answer regardless of the number of time integration steps taken, or the number of samples used. A solution to this is to supplement the Langevin updates with the Metropolis-Hastings (MH) Adjustment Langevin Algorithm (MALA), where MH acceptance guarantees detailed balance [Roberts and Tweedie, 1996]. A similar MH adjustment is also required for HMC [Duane et al., 1987].

The Langevin equation can be viewed as a particle implementation of the evolution of the particle probability density $q(\boldsymbol{x}(t))$, which is governed by the deterministic Fokker-Planck equation, a

continuity equation for the density,

$$\dot{q}(\boldsymbol{x}(t)) + \boldsymbol{\nabla} \cdot \boldsymbol{J} = 0, \quad \boldsymbol{J} = -q(\boldsymbol{x}(t))[\boldsymbol{\nabla}U(\boldsymbol{x}(t)) - \boldsymbol{\nabla}V(\boldsymbol{x}(t))] \equiv q(\boldsymbol{x}(t))\boldsymbol{v}. \quad (2)$$

We defined $V(\boldsymbol{x}(t)) = -\ln q(\boldsymbol{x}(t))$ and expressed current as density times velocity, where the two terms in the probability current $\boldsymbol{J}$ correspond to the two velocity terms in the Langevin equation. One can see from equation 2 that the stationary distribution $\dot{q}(\boldsymbol{x}, t) = 0$ is given by $p(\boldsymbol{x}|\boldsymbol{y}) = \exp(-U(\boldsymbol{x}))/p(\boldsymbol{y})$. The Fokker-Planck equation is a PDE, while the Langevin equation is stochastic (SDE). Langevin diffusion provides a particle description of the density $q(\boldsymbol{x}, t)$ and in the large time limit both equations lead to a stationary distribution equal to the posterior $p(\boldsymbol{x}|\boldsymbol{y})$.

If we replace the stochastic velocity in the Langevin equation 1 with the deterministic velocity in equation 2, we obtain the deterministic Langevin equation [Maoutsa et al., 2020, Song et al., 2020],

$$\dot{\boldsymbol{x}}(t) = \boldsymbol{v} = -\boldsymbol{\nabla}[U(\boldsymbol{x}(t)) - V(\boldsymbol{x}(t))]. \quad (3)$$

Equation 3 gives the dynamics of a particle representation of the density $q(\boldsymbol{x}(t))$, which in the large time limit $t \to \infty$ converges to the same distribution as the solution of equation 1. This follows from the fact that the stationary solution of equation 3, $\dot{\boldsymbol{x}}(t) = 0$ is given by $V(\boldsymbol{x}(t)) = U(\boldsymbol{x}(t)) + \text{c}$, with c being a constant independent of $\boldsymbol{x}$, which must thus be equal to $\log p(\boldsymbol{y})$, since $q(\boldsymbol{x}(t))$ is normalized. There is no stochastic noise in Equation 3, so it can be interpreted as a deterministic analog of the Langevin equation, which is a first order equation for velocity.

The main difficulty in solving equation 3 is the instantaneous density term $q(\boldsymbol{x}(t))$ needed to obtain $V(\boldsymbol{x}(t))$. Here we solve this by evolving $N$ particles together, using NFs to evaluate their particle density. While this makes the particles interacting, in the limit of large $N$ the particle evolution is independent. Furthermore, we can use the NF to do MH adjustment to correct for imperfect NF density estimation (Section 3.2). Upon time discretization of Equation 3 we obtain Algorithm 1. A detailed discussion of the algorithm implementation is given in Appendix B. We initialize at $t = 0$ by providing initial samples drawn at random from the known prior distribution $p(\boldsymbol{x})$, similar to Sequential Monte Carlo [Del Moral et al., 2006]. This has several positive features: since the posterior is proportional to the prior times the likelihood, the prior always covers the posterior. Moreover, when the likelihood is weakly informative the convergence of DLMC is fast. We then apply DLMC gradient updates to obtain the particle position updates. When initializing from the prior the initial density distribution cancels out the prior in the target, and the initial gradient update is the log likelihood term $\boldsymbol{\nabla}\mathcal{L}(\boldsymbol{x})$.

Equation 3 can be interpreted as a gradient based minimization of the time dependent objective $U(\boldsymbol{x}(t)) - V(\boldsymbol{x}(t))$. In this view we can optimize the objective using any optimization method. Initially, if we start the particles from the prior $p(\boldsymbol{x})$, DLMC is simply performing optimization of the target likelihood $p(\boldsymbol{y}|\boldsymbol{x})$. Later, as $q(\boldsymbol{x}(t)) \sim p(\boldsymbol{x}|\boldsymbol{y})$, the gradient of the optimization objective becomes zero, and the particles stop moving. One can use any optimization method to move the particles, so if the gradient of $U(\boldsymbol{x})$ is not available we can use gradient free optimization methods to do so. For gradient based optimization we can use many of the standard optimization methods [Nocedal and Wright, 2006], such as first order (stochastic or deterministic) gradient descent, momentum based methods, or (quasi) second order methods (e.g., L-BFGS, Newton, conjugate gradient etc.). The algorithm can be stopped when the particle distribution becomes stationary. For example, we can stop when the estimated first and second moments of the posterior stop changing.

Given the formulation above, the performance of DLMC depends on the ability of the NF to approximate the particle density and the corresponding gradient at each time step. The addition of MH updates (Section 3.2) help to correct for imperfect density estimation. However, in situations where the NF fit is a poor match for the target the MH acceptance can be very low. Furthermore, in high dimensions ($\mathcal{O}(10^3)$) we are limited by the training cost to use simple NF approximations. However, in this scenario DLMC can still be used to move particles quickly towards the typical set, acting as an initialization for MCMC in the NF latent space. This is discussed further in Section 5.5.

## 3 Normalizing Flow density estimation

Normalizing Flows provide a powerful framework for density estimation and sampling [Dinh et al., 2017, Papamakarios et al., 2017, Kingma and Dhariwal, 2018, Dai and Seljak, 2021]. These models map the $d$-dimensional data $\boldsymbol{x}$ to $d$-dimensional latent variables $\boldsymbol{z}$ through a sequence of invertible

transformations $f = f_1 \circ f_2 \circ ... \circ f_L$, such that $\boldsymbol{z} = f(\boldsymbol{x})$ and $\boldsymbol{z}$ is mapped to a base distribution $\pi(\boldsymbol{z})$, which is chosen to be a standard Normal distribution $N(0, \boldsymbol{I})$. The probability density of data $\boldsymbol{x}$ can be evaluated using the change of variables formula:

$$q(\boldsymbol{x}) \equiv e^{-V(\boldsymbol{x})} = \pi(f(\boldsymbol{x})) \left| \det \left( \frac{\partial f(\boldsymbol{x})}{\partial \boldsymbol{x}} \right) \right| = \pi(f(\boldsymbol{x})) \prod_{l=1}^{L} \left| \det \left( \frac{\partial f_l(\boldsymbol{x})}{\partial \boldsymbol{x}} \right) \right|. \tag{4}$$

The Jacobian determinant of each transform $J_l = |\det(\frac{\partial f_l(\boldsymbol{x})}{\partial \boldsymbol{x}})|$ must be easy to compute for evaluating the density, and the transformation $f_l$ should be easy to invert for efficient sampling. In this paper we use Sliced Iterative Normalizing Flow (SINF) [Dai and Seljak, 2021], which has been shown to achieve better performance for small training data (below a few thousand particles), and is considerably faster, than alternatives.

## 3.1 Normalizing Flow as a preconditioner

While the core of DLMC is the NF based density gradient, the same NF can be used for additional convergence acceleration towards the target. The proposed algorithm can be slow when the condition number of the problem is high. A typical solution is to precondition with a covariance matrix, which mimics the global curvature of the problem. This becomes more difficult if the curvature is position dependent. NFs can address this problem by warping the space in which we perform the optimization. This warping maps spatially varying curvature to the latent space where the curvature is constant and isotropic [Parno and Marzouk, 2018, Hoffman et al., 2019]. We can implement DLMC in the NF latent space $\boldsymbol{z}$, where the target distribution $p(\boldsymbol{x}|\boldsymbol{y})$ is transformed into

$$p(\boldsymbol{z}|\boldsymbol{y}) = p(\boldsymbol{f}^{-1}(\boldsymbol{z})|\boldsymbol{y})J(\boldsymbol{z}), \tag{5}$$

where $J(\boldsymbol{z}) = \left| \det \left( \frac{\partial \boldsymbol{f}^{-1}(\boldsymbol{z})}{\partial \boldsymbol{z}} \right) \right|$ is the absolute value of the Jacobian determinant. We can define the potential in latent space as

$$U(\boldsymbol{z}) = U(\boldsymbol{f}^{-1}(\boldsymbol{z})) - \ln J(\boldsymbol{z}). \tag{6}$$

If we can evaluate $\boldsymbol{\nabla}U(\boldsymbol{x})$ using methods such as auto-differentiation and if the NF Jacobian is also differentiable we can also evaluate the gradient in latent space $\boldsymbol{z}$ as

$$\boldsymbol{\nabla}U(\boldsymbol{z}) = \boldsymbol{\nabla}U(\boldsymbol{x})\frac{d\boldsymbol{x}}{d\boldsymbol{z}} - \boldsymbol{\nabla}\ln J(\boldsymbol{z}) \tag{7}$$

In latent space the latent variables are distributed as $\pi(\boldsymbol{z}) = \exp(-V(\boldsymbol{z})) = N(0, \boldsymbol{I})$, so $V(\boldsymbol{z}) = \boldsymbol{z}^T\boldsymbol{z}/2 + (d/2)\ln(2\pi)$ and $\boldsymbol{\nabla}V(\boldsymbol{z}) = \boldsymbol{z}$. This gives the deterministic Langevin gradient descent updates as

$$\boldsymbol{z}(t + \Delta t) = \boldsymbol{z}(t) - \boldsymbol{\nabla}U(\boldsymbol{z}(t))\Delta t + \boldsymbol{z}(t)\Delta t. \tag{8}$$

It is interesting to compare this deterministic update to the standard stochastic Langevin equation update, which in latent space is

$$\boldsymbol{z}(t + \Delta t) = \boldsymbol{z}(t) - \boldsymbol{\nabla}U(\boldsymbol{z}(t))\Delta t + \boldsymbol{\eta}', \tag{9}$$

where $\boldsymbol{\eta}' \sim N(0, 2\boldsymbol{I}\Delta t)$. Both updates have the potential gradient term, which is the deterministic drift velocity. The difference between the two is that DLMC also uses the current position of the sample $\boldsymbol{z}$ for a deterministic update of the density gradient $\boldsymbol{z}(t)\Delta t$, which moves the particles radially outwards from the center of the latent space, while the stochastic Langevin algorithm adds a Gaussian random variable $\boldsymbol{\eta}'$ to the update. We find that performing DLMC updates in the NF latent space accelerates convergence.

## 3.2 Normalizing Flow for Metropolis-Hastings adjustment

The main requirement of DLMC is that $q(\boldsymbol{x}(t))$ properly describes the density of samples $\boldsymbol{x}(t)$ at time $t$. This can only be ensured in the large $N$ limit. Moreover, for non-convex optimization problems it is possible that the samples get stuck at local minima far from the global minimum. One can add a Metropolis-Hastings (MH) adjustment to correct for imperfect density estimation,

---

**Algorithm 1** Deterministic Langevin Monte Carlo with Normalizing Flows

---

1: **Input:** initial samples $\boldsymbol{x}^i(t=0)$ drawn at random from prior $p(\boldsymbol{x})$, size $N$, access to $\mathcal{L}(\boldsymbol{x}) + \mathcal{P}(\boldsymbol{x}) = U(\boldsymbol{x})$ and its gradient.
2: Initial update:
3: **for** $i = 1$ to $N$ **do**
4:     $\boldsymbol{x}^i(\Delta t) = \boldsymbol{x}^i(t=0) - \boldsymbol{\nabla}\mathcal{L}(\boldsymbol{x}^i(t))\Delta t$
5: **end for**
6: **while** not converged **do**
7:     $t \leftarrow t + \Delta t$
8:     Run Normalizing Flow on current particle positions to obtain $q(\boldsymbol{x}(t)) = \exp(-V(\boldsymbol{x}(t)))$ and map $\boldsymbol{z}(t) = f(\boldsymbol{x}(t))$
9:     **for** $i = 1$ to $N$ **do**
10:         **if** latent space update **then**
11:             Map to latent space: $\boldsymbol{z}^i(t) = f(\boldsymbol{x}^i(t))$
12:             DL update: $\boldsymbol{z}^i(t+\Delta t) = \boldsymbol{z}^i(t) - [\boldsymbol{\nabla}U(\boldsymbol{z}^i(t)) - \boldsymbol{z}^i(t)]\Delta t$
13:             Inverse map: $\boldsymbol{x}^i(t+\Delta t) = f^{-1}(\boldsymbol{z}^i(t+\Delta t))$
14:         **else**
15:             DL update: $\boldsymbol{x}^i(t+\Delta t) = \boldsymbol{x}^i(t) - \boldsymbol{\nabla}[U(\boldsymbol{x}^i(t)) - V(\boldsymbol{x}^i(t))]\Delta t$
16:         **end if**
17:         MH update: draw a sample $\tilde{\boldsymbol{x}}^i \sim q(\boldsymbol{x}(t))$, replace $\boldsymbol{x}^i$ with $\tilde{\boldsymbol{x}}^i$ with probability $r(\boldsymbol{x}^i, \tilde{\boldsymbol{x}}^i)$
18:     **end for**
19: **end while**
20: **Output**: a set of particles $\boldsymbol{x}^i(t)$ distributed as a target distribution $p(\boldsymbol{x}|\boldsymbol{y})$.

---

eliminating samples that get stuck far from the global minimum. This step is particularly necessary for disconnected multi-modal peaks of the target to be properly equilibrated. We note that whilst LMC (i.e., MALA) and HMC utilize local MH adjustment, they typically cannot equilibrate between separate posterior peaks without the use of methods such as annealing.

Specifically, at a given $t$ we can draw independent particles $\tilde{\boldsymbol{x}}^i \sim q(\boldsymbol{x}(t))$, one for each existing particle $\boldsymbol{x}^i$. We compare the new particle against an existing particle to decide whether we accept or reject it. Since these new samples are independent the MH acceptance rate is (e.g., Albergo et al. [2019])

$$r(\boldsymbol{x}, \tilde{\boldsymbol{x}}) = \min\left\{1, \frac{p(\tilde{\boldsymbol{x}}, \boldsymbol{y})q(\boldsymbol{x})}{p(\boldsymbol{x}, \boldsymbol{y})q(\tilde{\boldsymbol{x}})}\right\}. \tag{10}$$

Note that the normalization constant cancels out in this ratio, so we can evaluate this with knowledge of $U(\boldsymbol{x})$ only.

Initially, when $q(\boldsymbol{x}(t))$ is very broad compared to $p(\boldsymbol{x}|\boldsymbol{y})$, the acceptance rate of MH will be low, since most of the samples drawn will have low $p(\boldsymbol{x}|\boldsymbol{y})$, so this process may only eliminate the worst performers, and replace them with samples in the higher density region of $p(\boldsymbol{x}|\boldsymbol{y})$. As DLMC progresses $q(\boldsymbol{x}(t))$ becomes closer to $p(\boldsymbol{x}|\boldsymbol{y})$, and the acceptance increases and reaches $r(\boldsymbol{x}, \tilde{\boldsymbol{x}}) = 1$ if the NF learns the target perfectly. In high dimensions the MH acceptance can remain low. However, the MH update is still necessary for proper equilibration of multi-modal peaks.

## 4 Related work

NFs and transport maps for Bayesian posterior analysis have been explored in Parno and Marzouk [2018], Hoffman et al. [2019], Arbel et al. [2021]. These approaches use NFs as latent space preconditioners, using standard MCMC methods such as HMC, LMC or MH deployed in latent space, where the geometry may be more favorable for fast mixing. In Albergo et al. [2019] NFs are used as an MH transition proposal, giving independent samples. DLMC goes beyond these NF applications in that it also uses NFs for the density gradient in the deterministic Langevin equation.

DLMC is an ensemble method, in that all of the particles inform the NF density evaluation, which is in turn used to update each particle position. There have been many other ensemble methods proposed in the literature, such as the Affine Invariant Ensemble Sampler [Goodman and Weare, 2010] or Differential Evolution MCMC [Ter Braak, 2006]. DLMC differs from these methods in

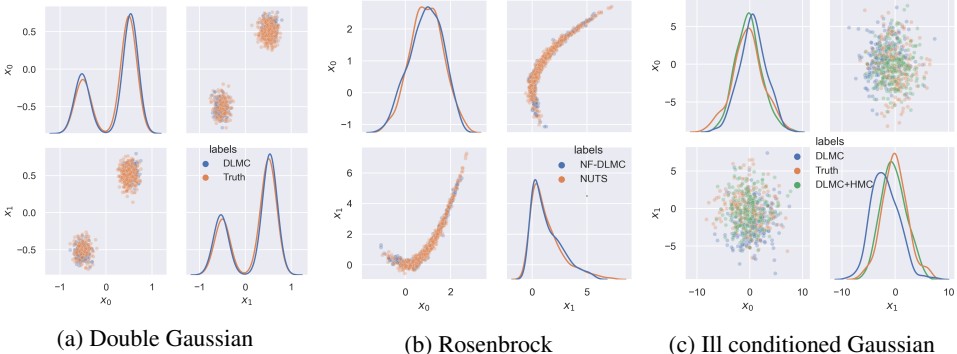

(a) Double Gaussian         (b) Rosenbrock         (c) Ill conditioned Gaussian

Figure 1: Panel (a): results on $d = 100$ Gaussian mixture after 90 DLMC iterations (blue), compared to the true distribution (orange). We show 1d density projections and 2d particle distributions along the first two parameters. Panel (b): same for Rosenbrock function in $d = 32$ after 50 DLMC iterations. Panel (c): particle distributions for the $d = 1000$ ill conditioned Gaussian after 28 DLMC iterations, along with the distribution following 5 subsequent latent space HMC iterations (green). The Gaussian mixture uses 500 particles, the Rosenbrock uses 1000, and the ill conditioned Gaussian uses 2000.

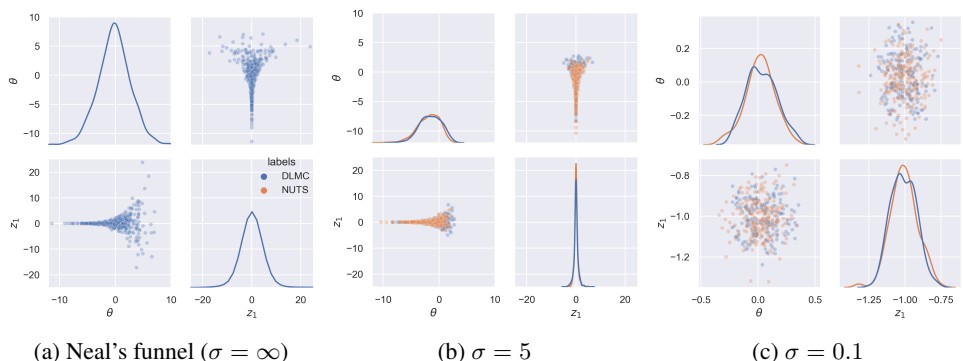

(a) Neal's funnel ($\sigma = \infty$)        (b) $\sigma = 5$        (c) $\sigma = 0.1$

Figure 2: DLMC results on $d = 101$ hierarchical variance problem (funnel problem), compared to a very long NUTS run. We show posteriors of $\theta$ and $z_1$. Left panel shows Neal's funnel: it corresponds to noise $\sigma \to \infty$, so no updates are needed since the likelihood update is zero. Middle panel shows $\sigma = 5$: despite the large noise, the $d = 100$ latent variables are informative of $\sigma$ and the posterior is narrower than the prior. Right panel shows $\sigma = 0.1$: here the posterior is narrowed further, and is close to a Gaussian.

that it evaluates the particle density explicitly, and uses the density and target gradient to update the particle evolution.

Two deterministic algorithms related to DLMC that have recently been proposed are Stein Variational Gradient Descent (SVGD) [Liu and Wang, 2016], which uses direct pair summation over all particles, and score based interacting particles [Maoutsa et al., 2020], inspired by generative score based models [Song et al., 2020], which use the particles to directly estimate the density gradient term. DLMC differs from these in using NFs instead of direct pair interactions, which we find requires fewer particles to accurately represent the particle density.

DLMC is related to Variational Inference (e.g., Blei et al. [2017]) in that both approximate the posterior with a parametrized normalized function $q(\boldsymbol{x})$. However, DLMC does not use standard variational optimization such as KL minimization to optimize $q(\boldsymbol{x})$, and does not use $q(\boldsymbol{x})$ for the posterior, which is reported by the particles instead. In our experiments we observe that the posterior distribution of DLMC particles agrees better with the exact posteriors than the samples from the NF density. This is expected, since we use the MH step to correct the NF. DLMC thus contains components of both sampling and optimization: sampling as optimization in the space of distributions has been pursued in many directions since Jordan et al. [1998]).

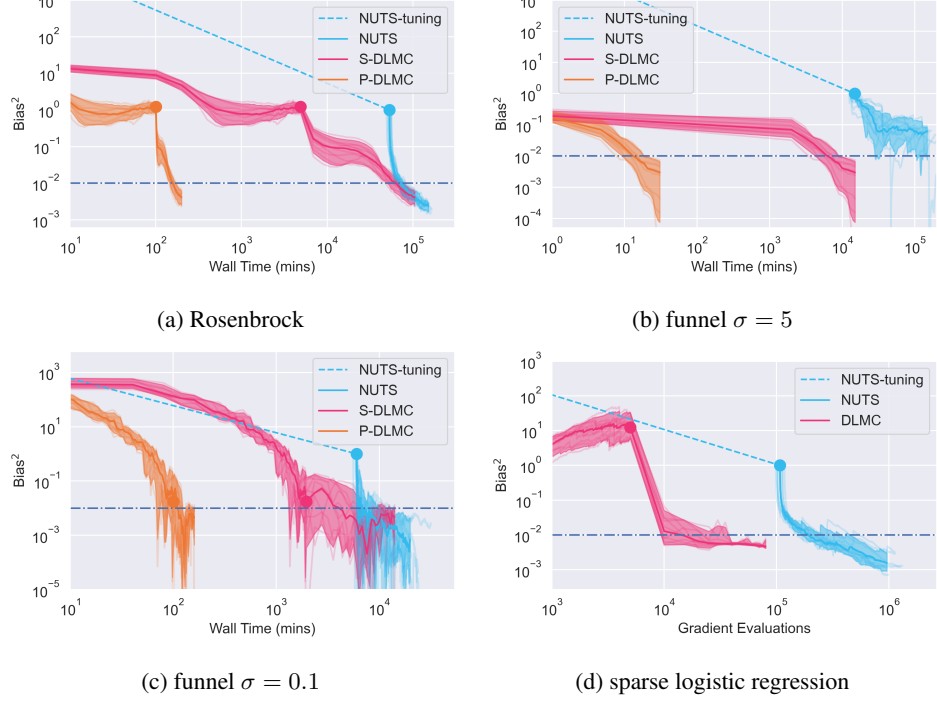

(a) Rosenbrock

(b) funnel $\sigma = 5$

(c) funnel $\sigma = 0.1$

(d) sparse logistic regression

Figure 3: Bias-squared on the second moment for the $d = 32$ Rosenbrock function (panel (a)), $d = 101$ hierarchical funnel vs wall clock time (panels (b) and (c)) and for $d = 51$ sparse logistic regression versus number of likelihood gradient evaluations (panel (d)). In panels (a)-(c) we show serial (S-DLMC) and parallel (P-DLMC) versions. Solid circles denote the end of any burn-in phases.

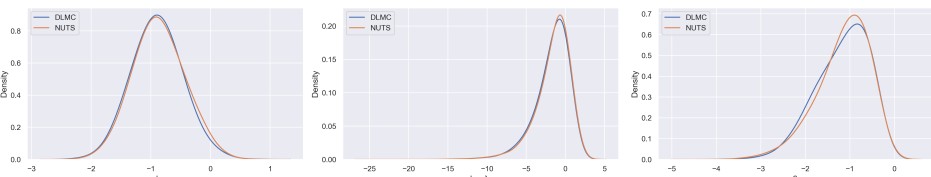

Figure 4: Results on $d = 51$ sparse logistic regression after 100 iterations (blue), compared to the distribution obtained from a very long NUTS run (orange). We show three representative variables, showing good agreement. The corresponding bias-squared is 0.01 (figure 4).

## 5   Examples

The main DLMC application is for expensive likelihoods. We can simulate this by synthetically creating expensive model evaluations, but will also compare the performance in terms of the number of likelihood evaluations. The implementation of DLMC requires NF and particle evolution. A discussion of implementation details can be found in Appendix B. For NF we use SINF, which has very few hyper-parameters [Dai and Seljak, 2021], is fast, and iterative. The number of layers $L$ can be chosen based on cross-validation, where we set aside 20% of the samples, and iterate until validation data start to diverge from the training data. However, for the $d = 1000$ Gaussian (Section 5.5) we fix $L = 5$. Typical SINF training time is of order seconds on a CPU.

The implementation of particle evolution requires us to specify the learning rate for the particle updates. At each iteration, we take Adagrad updates in the $\boldsymbol{\nabla}(U(\boldsymbol{x}(t)) - V(\boldsymbol{x}(t)))$ direction [Duchi et al., 2011]. We use learning rates between $0.001$ and $0.1$, with smaller learning rates being more robust for targets with complicated geometries such as funnel distributions.

In our plots we focus on the quality of marginal posteriors, because that is typically the primary goal of Bayesian inference. To quantify the quality of the posteriors numerically we follow Hoffman et al.

[2019] and show the squared bias of the second moment versus wall clock time or number of gradient evaluations. The precision of this number is limited by the effective number of samples (ESS). Here we choose bias-squared of 0.01 as sufficient in terms of posteriors, and corresponds for a Gaussian distribution to an ESS of 200. The number of particles $N$ is a hyperparameter that must exceed this ESS. Overall we find there is a tradeoff between number of particles and number of iterations, such that starting with a larger number of particles does not always imply a higher overall computational cost, due to the improved NF density estimation resulting in fewer DLMC iterations.

The main baseline we compare against is the No-U-Turn Sampler (NUTS) [Hoffman et al., 2014], an adaptive HMC variant implemented in the NumPyro library [Phan et al., 2019]. It requires tuning and burn-in, which need to be included in the computational budget. NUTS is a standard baseline since it typically outperforms other samplers such as LMC, SVGD and SMC. Where we include NUTS as a baseline, we use 500 tuning steps. NUTS has the number of leapfrog steps as a tunable parameter, and when it is one HMC becomes LMC/MALA [Girolami and Calderhead, 2011].

In some of our experiments we compare default DLMC (denoted serial, S-DLMC) to parallel DLMC (P-DLMC), where for the latter we show the wall clock time based on the assumption that we can evaluate the likelihood gradient in parallel on $N$ CPU cores. We assume a likelihood gradient cost of 1 minute, and the cost of the NF itself (seconds) is negligible.

## 5.1 Gaussian mixture

Multimodal distributions are a failure mode for many standard MCMC samplers as they cannot move from one peak to another when the potential barrier between them is too high. They need to be augmented with an annealing procedure, where one slowly morphs the target distribution from the prior to the posterior. Annealed Importance Sampling [Neal, 2001] and Sequential MC (SMC) [Del Moral et al., 2006] are examples of such procedures. They require equilibrating the distribution at every temperature level using standard MCMC, and as a result are significantly slower than standard MCMC. DLMC uses a different strategy from annealing, but can also handle multi-modal distributions, as long as an NF can approximate it. As an example we take a Gaussian mixture with 1/3 of the posterior mass in the first peak and 2/3 in the second peak, in $d = 100$.

We choose $N = 500$, drawn from a broad initial prior $(-2, 2)^d$. The log of the prior to posterior volume ratio is 415, which is a challenge for non-gradient based MCMCs, as they become very slow in such settings. We apply logit transforms to all the variables such that we sample in an unconstrained space. DLMC is gradient based and makes rapid progress towards the region of high posterior mass. The resulting DLMC posterior after 90 iterations for the first two variables is shown in figure 1, and is in a near perfect agreement with the true distribution.

## 5.2 Rosenbrock function

The 32-d example comes from a popular variant of the multidimensional Rosenbrock function [Rosenbrock, 1960], which is composed of 16 uncorrelated 2-d bananas. The model log-likelihood is $\ln L = -\sum_{i=1}^{n/2} \left[ (g(x_{2i-1})^2 - x_{2i})^2/Q + (x_{2i-1} - 1)^2 \right], Q = 0.1, n = 32$, with Gaussian prior $N(0, 6^2)$ on all the parameters. We assume $g(x) = x$ and use an initial burn-in phase with $N = 10$ for 50 iterations, before upsampling to $N = 1000$. The posterior for the first two variables is shown in figure 1. We see that DLMC perfectly matches NUTS. The quality of the posterior versus wall clock time is shown in figure 3, where we are idealizing a data assimilation process by assigning a high computational cost of 1 minute to the evaluation of $g(x)$. This allows us to compare serial and parallel DLMC. S-DLMC achieves results that are competitive with NUTS. P-DLMC can improve the wall clock time by $N = 1000$.

## 5.3 Hierarchical Bayesian analysis

Hierarchical Bayesian analyses are very common in many applications. They often lead to funnel type posteriors, which are challenging for standard samplers. Moreover, forward models relating the variance of the latent space to some parameters of interest can be expensive. As an example, in cosmology data analysis applications we measure Fourier modes and want to determine their power spectrum, which in turn constrains cosmological parameters. The evaluation of the power spectrum as a function of cosmological parameters requires an expensive PDE evolution of structure in the

universe as a function of time [Lewis et al., 2000]. Here we choose a simplified version where the power variance $P = g(\theta)$ is determined by cosmological parameter $\theta$ and controls the variance of parameters $z_i$. Parameters $z_i$ are not directly observable, and instead we observe their noisy version with a likelihood of the data $\boldsymbol{y}$, so a combined prior and likelihood are

$$p(\theta) = N(0, 3^2), \ p(z_i|\theta) = N(0, P = g(\theta)), \ p(y_i|z_i, \theta) = N(z_i, \sigma^2), \ i = 1, \ldots, d-1, \quad (11)$$

where $\sigma^2$ is the measurement noise and $g(\theta)$ is an expensive function that evaluates the power spectrum $P$ as a function of $\theta$. To connect to Neal's funnel we assume the result of the PDE is that the power spectrum depends exponentially on $\theta$, with an approximate relation $P = g(\theta) = \exp(\theta)$, with a 1 minute computational cost of performing $g(\theta)$. The goal is the posterior of parameter $\theta$ marginalized over $z_i$. We use $N = 500$.

In the limit $\sigma^2 \to \infty$ the setup corresponds to Neal's funnel, which is deemed problematic for many samplers such as HMC [Neal et al., 2011]. Note however that the DLMC solution in the $\sigma^2 \to \infty$ is trivial: the data likelihood is non-informative, and after drawing samples from the prior of equation 11 the likelihood updates of equation 11 are zero, so the DLMC stopping criterion (stationary posterior) is satisfied immediately.

To make the problem closer to actual applications in cosmology we analyze it with a finite noise $\sigma^2$, such that the data are informative of the parameter $\theta$. We create a simulated data set with the true value $\theta_{\text{true}} = 0$ and use $d = 101$. The resulting posterior for $\sigma = 5$ and $\sigma = 0.1$ is compared to NUTS in figure 2. We also show the prior (i.e., Neal's funnel) to highlight that the posterior is narrow compared to the prior. For $\sigma = 5$ we used 15,000 likelihood calls, which is orders of magnitude fewer likelihood evaluations than the NUTS baseline, which struggles to converge at all (figure 3) without the help of a reparametrization or preconditioner [Hoffman et al., 2019].

To highlight the advantage of DLMC with parallelization we further show the performance of (embarrassingly) parellel DLMC, where we evaluate the likelihood update for all 500 particles in parallel, so each iteration is $\sim 1$ minute. This converges to the required precision $b^2 = 10^{-2}$ in 30 minutes. Sequential MCMC methods such as NUTS cannot be so efficiently parallelized because they require a hyperparameter tuning and burn-in phase (each of order $10^4$ likelihood calls). This leads to 3-4 orders of magnitude larger wall clock time. SMC can take advantage of parallelization, but needs a lot of temperature levels and a lot of steps to equilibrate at each temperature level [Del Moral et al., 2006, Wu et al., 2017], such that the overall wall clock time is still orders of magnitude larger than parallel DLMC.

## 5.4 Sparse logistic regression

Hierarchical logistic regression with a sparse prior applied to the German credit dataset is a popular benchmark for sampling methods [Dua and Graff, 2017]. We use the example from Hoffman et al. [2019] with $d = 51$, applying a log-transform to bounded variables such that we sample in an unconstrained space. This is a challenging example for NUTS, with samples requiring on average $\sim 125$ leapfrog steps.

We run an extended burn-in phase with 10 particles to shrink from the prior to posterior volume, using 500 iterations. We then upsample to 5000 samples, with DLMC converging to the correct solution within 10 iterations. Upsampling to lower sample sizes, such as 1000 particles, also converges to the correct solution, but requires more DLMC iterations such that the computational cost is not reduced. The resulting marginals are shown in figure 4 for three representative variables showing good agreement.

## 5.5 Ill conditioned Gaussian

To asses the performance of DLMC in high dimensions we consider an ill conditioned Gaussian target in $d = 1000$. We construct the target covariance $\Sigma$ by sampling its eigenvalues from $\mathrm{Gamma}(1, 1)$, giving a condition number of $\sim 4000$. The prior and likelihood are then defined as

$$p(\boldsymbol{x}) = N(0, \Sigma_\pi = 100^2 I), \ p(\boldsymbol{y}|\boldsymbol{x}) = N(\boldsymbol{x}, \Sigma_L = (\Sigma^{-1} - \Sigma_\pi^{-1})^{-1}), \quad (12)$$

where the prior and likelihood covariance are chosen such that the posterior covariance is $\Sigma$.

In this high dimensional setting we are limited by the computational cost of the NF, and the ability of the NF to approximate the current particle density. However, we can use a simple flow with 5 layers,

to move particles quickly towards the typical set with DLMC. In this experiment we evolved 2000 particles until we satisfied the condition $\mathcal{V} = \sum_{i=1}^{d} \langle \boldsymbol{x} \cdot \boldsymbol{\nabla} U(\boldsymbol{x}) \rangle_i \leq 1.1d$, where the sum is over the target dimensions. This virial condition is chosen because initially we have $\mathcal{V} \gg d$, with the particle ensemble satisfying $\mathcal{V} = d$ at equilibrium [Leimkuhler and Matthews, 2015].

For this problem we are unable to achieve low bias with DLMC alone. However, we can treat DLMC as an initialization for HMC i.e., we run DLMC until we reach the virial threshold and then perform parallel HMC with each particle in the NF latent space. DLMC reaches the virial threshold in 28 steps. Running parallel HMC chains in latent space with 10 leapfrog steps and the step size targeting an acceptance rate of 0.65, we reach a bias-squared of 0.01 after 5 HMC iterations. This corresponds to $\sim 10^5$ serial likelihood evaluations, and $\sim 100$ parallel evaluations. The resulting particle distributions from DLMC and the subsequent latent space HMC are shown in figure 1. By comparison, NUTS requires $\sim 10^5$ serial likelihood evaluations to reach a bias-squared of 0.01. For this example the serial cost of DLMC+HMC and NUTS is comparable, with DLMC+HMC providing a performance improvement of $\sim 10^3$ when implemented in parallel. In this experiment we have used a very simple implementation of HMC, and the convergence rate could likely be improved by using some ensemble adaptation scheme e.g., Hoffman and Sountsov [2022].

### 5.6 Ablation studies and comparison to particle deterministic methods

In Appendix C we compare to kernel density estimation based DLMC and SVGD, both of which give inferior results, confirming that the NF is the key ingredient for the success of deterministic methods.

In Appendix D we present results of ablation studies. We show DLMC without MH, DLMC without NF preconditioning, and pure NF MH without DLMC. In all cases the results are inferior to DLMC with preconditioning and with MH adjustment. We also show NF with MH without DL is inferior to DLMC.

## 6 Conclusions

We present a new general purpose Bayesian inference algorithm we call Deterministic Langevin Monte Carlo with Normalizing Flows. The method uses the deterministic Langevin equation, which requires the density gradient, which we propose to be evaluated using a Normalizing Flow determined by the positions of all the particles at a given time. Particle positions are then updated in the direction of log target density minus log current density gradient. To correct for imperfect NF density estimation, we add an MH step. The process is repeated until the particle distribution becomes stationary, which happens when the particle density equals the target density.

In our experiments we find that DLMC significantly outperforms baselines such as LMC, HMC and SVGD, and even more in terms of wall-clock time when likelihoods can be evaluated in parallel on a multi-core computer. A possible limitation is that NF training may not learn the true target distribution for small $N$. While DLMC with MH does offer some robustness against imperfect density estimation, it can fail in situations where the MH acceptance rate is very low, such that it cannot reach the stationary distribution in a finite number of steps. This is particularly apparent in $\mathcal{O}(10^3)$ dimensions, where we are limited by the NF training cost and the fidelity of its gradient approximation. Here, convergence on the correct target can be achieved by running DLMC until the particle ensemble reaches the typical set, at which point we perform a short MCMC run in latent space. Indeed, treating DLMC as an initialization for a latent space MCMC in this way can provide some additional robustness through the associated asymptotic guarantees, whilst avoiding the expensive tuning and burn-in process typically required for MCMC. It would nonetheless be interesting to explore what NF architectures, regularization, and training procedures safeguard against low MH acceptance rates.

## Acknowledgements

This material is based upon work supported by the U.S. Department of Energy, Office of Science, Office of Advanced Scientific Computing Research under Contract No. DE-AC02-05CH11231 at Lawrence Berkeley National Laboratory to enable research for Data-intensive Machine Learning and Analysis. We thank Minas Karamanis, David Nabergoj and James Sullivan for useful discussions.

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
