# OpenReview forum: "Deterministic Langevin Monte Carlo with Normalizing Flows for Bayesian Inference"
_NeurIPS.cc/2022/Conference — NeurIPS 2022 Accept_

### Official Review · Reviewer_yr5Z · 2022-07-10

**Rating:** 4
**Confidence:** 4
**Soundness:** 3 good
**Presentation:** 2 fair
**Contribution:** 2 fair

**Summary:**

The authors of this work propose using the deterministic “dual” of the langevin equation for simulating from a target posterior. This dual presents difficulties since it requires estimating the score of the FPK equations solutions, to do this the authors propose using normalizing flows, furthermore , the authors “precondition” the flows in latent space via simulating what seems to be latent Gaussian dynamics at equilibrium, this “preconditioning” improves/stabilizes the overall procedure.

**Questions:**

Comments/Suggestions:

1. The deterministic Langevin Equation is well known and used in the diffusion score matching literature [1] (see Eq X) and should be acknowledged appropriately and maybe briefly discussed. Notice the velocity term coincides with the score.
2. Equation 2 is broken U(x) should be U(x(t)) in order to be consistent, however this is also wrong, the FPK equation is defined pointwise for all x \in R^d  that is its defined as q(x,t) or q_t(x) , it's not only defined pathwise on x(t), this notation is inaccurate. The f(x(t)) notation should be reserved for the deterministic langevin equation, but the FPK equation is defined pointwise. Across the whole paper the usage of q(x), q(x(t)) and x vs x(t) in general is quite inconsistent.
3. From flows setup you would expect p(z|y) to be Gaussian but this is never made clear, instead you refer to \pi(z)  being Gaussian which was never explicitly defined in the context of p(x|y) ? let's assume that in your notation \pi(z) = p(z|y) = N(0,1) then indeed the score of q_t(z) is infact -z (i.e. V(z)=z) however this implies that z(t) evolves entirely according to linear Gaussian dynamics , not only that but if the score is indeed -z this implies that q_t(z) = N(z; 0,1) which is at equilibrium at does not depend on time whatsoever. What is the point of simulating gaussian noise at equilibrium in latent space ?  Formally I cannot see how this can obtain finite time estimates of q_t(x)  (t< \infty)
4. To clarify the previous point a bit further if indeed V(z)=z then this implies directly that -ln q_t = || z||^2/2 which implies directly that q_t is an independent Gaussian where q_t(z) solves the FPK equation for the density implied by the ODE dynamics in Eq 8. Remember U(z) and V(z) cannot be chosen independently, given U(z)  then grad(-U(z)) implies an FPK equation  where V(z) depends on the solution to that equation. One can't just go  oh let V(x) = ln Gaussian without first proving that q_t(z) = Gaussian.
5. Continuing the line of reasoning dz_t =grad(-U(z))dt +dW_t should have as its steady state N(0,1)  if z_0 where samples from \pi(z)=p(z|y) then in fact V(z) = z can easily be derived, but again you'd be just simulating an OU process at equilibrium … What is the advantage of this? Can you formally prove an advantage of doing this? Or at least motivate it conceptually ?


Questions on methodology:

1. In your algorithm you don't have any pseudocode for the z(t) dynamics ? Why is this ? I found this quite confusing.
2. For estimating q_t(x) you just say run normalizing flows  on x(t) ? so here you just do density estimation with normalizing flows correct ? (you learn flows in the standard way) . It really feels this is the essence to the method, all that is being done is running normalizing flows to estimate q_t(x) and then V(x) and then some adhoc stabilizing noise procedure is done with a lack of motivation.
3. How is the preconditioning used then ? you do some extra gaussian dynamics in latent space with the learned flows to stabilize things a bit ?  (Please clarify and give more motivation)
4. I can't seem to find an ablation for the effect of preconditioning/simulating the Gaussian, what effect does the adhoc preconditioning routine have? You can still use NF to fit the score without this step.


**Limitations:**

Overall there are clear counter points in this work that lack detailed discussion:

1. When going from Langevin to the deterministic version, more approximation is required, in  sense it is a harder problem, whilst this allows for a hybrid version between ULA / normalizing flows, the authors do not motivate well why one would want to pursue such a problem (which requires more difficult approximation schemes).
2. The whole derivation of the latent dynamics in the flow/gaussian space seems lacking and also is motivated. At the end of the day the equations you have seem to simulate Gaussian dynamics at equilibrium, this is either not particularly useful or wrong (if somehow they are out of equilibrium in practice).



[1] Song, Y., Sohl-Dickstein, J., Kingma, D.P., Kumar, A., Ermon, S. and Poole, B., 2020. Score-based generative modeling through stochastic differential equations. arXiv preprint arXiv:2011.13456.

**Strengths And Weaknesses:**

Pros:
1. A sound methodology for simulating the deterministic “dual” of the Langevin equation using normalizing flows.
2. Sound and successful experimental procedure.

Cons:

1. The overall guarantees of the method (theoretically) are not clear and there's no acknowledgement if the method is mostly heuristic.
2. Furthermore, intuition behind the method's preconditioning step is also not presented clearly. There are some potential conceptual/theoretical issues illustrated by the reviewer in point 3 (with a potential contradiction).
3. Lack of motivation when moving from langevin to its stochastic counterpart.

---

> ### Author Response · Authors · 2022-08-01
> **DLMC is deterministic, with sound theoretical motivation, which is confirmed by numerical results; latent space DLMC is just DLMC in a different basis**
>
> Q1: we cite Song et al several times (ref 6), including in key equation 3. Moreover, Song et al. is preceeded by ref 3 Maoutsa et al (as acknowledged in Song et al), which we refer to and compare against in introduction and in appendix A.
>
> Q2: we have dropped the time index, ie we wrote x instead of x(t), in places where we assumed this will not cause confusion. In this we have followed Maoutsa et al (ref 3), where it is also dropped between their eq 4 and 5. We refer to that paper for an alternative derivation that demonstrates the validity of our equations. We will clarify and improve upon our time notation.
>
> Q3-5: the reviewer has concerns about the preconditioning with NF. We refer to Hoffman et al (ref 14) for a more detailed motivation and
> discussion of the method. To summarize it: one can work in any parameter basis using a bijective map of parameters from x to z. One very simple basis is NF latent space, where q_t(z) is N(0,1), and grad V(z)=z. The target p(x|y) and U(x) is also transformed to p(z|y) and U(z) (eq 6). There are no approximations or assumptions in this procedure, and there is nothing ad-hoc about it: if one accepts the validity of a single step DLMC update in space x then the validity of a single step DLMC update in space z follows immediately, it is just a parameter transformation.
>
> As for motivation, it is the same motivation as with any preconditioning, which is to reduce the condition number of gradient based methods: for high condition number problems gradient based optimization is slow because of zig-zagging along the narrow potential ravine. Second order methods improve upon this by using inverse Hessian information. A generalization of second order method is to allow spatially varying Hessian. NFs can thus be viewed as powerful preconditioners that go beyond Hessian type preconditioning and allow spatially varying Hessian for an even faster convergence of gradient based optimization.
>
> Contrary to reviewer statement p(z|y) is NOT Gaussian N(0,I) until we converge, ie at the end of DLMC. Prior to that it is whatever eq 5 gives for it.
>
> There is no stochastic noise in our procedure (it is deterministic), so we are unclear why the reviewer talks about stochastic noise, simulating Gaussian noise, stabilizing noise procedure, writes down Wiener noise term dW_t etc. None of this is relevant for what we do. The reviewer is correct that NF is the essence of our method, as we state in introduction, but there is no adhoc noise stabilizing procedure in our method.
>
> It is perhaps worth pointing out that the NF changes at every iteration, ie q_t(x) depends on t since particles move. So while q_t(z)=N(0,I) (in the infinite N limit and assuming NF is a universal approximator that has converged) for any time t, grad U(z) is changing with time since the NF map is changing and thus the Jacobian of NF J=|dx/dz| is changing in time. So eq. 8 should not be viewed as some fundamental
> equation to be solved on its own, but rather a rewrite of equation 3 in the NF latent space basis, which changes with every iteration step.
> We do this so that the gradient descent is accelerated because the condition number has been reduced.
>
> We did not include algorithm for latent space because it is a simple replacement of equation 3 with eq 7 and 8. We will include it in revised version, as it is a one line change of the Algorithm 1.
>
> We do perform ablation study of the impact of this preconditioning, this is shown in fig 8 where we show convergence is faster in NF latent space. As the reviewer correctly points out one can implement DLMC without this latent space preconditioning, but we find the convergence is faster with this step.
>
> When going from the Langevin to the deterministic version we trade the stochastic nature of Langevin MC, which is a random walk and thus slow, for the deterministic density term, which converges faster, but relies on our ability to determine the density from the particles accurately. This is a well defined trade-off and in our examples we show DLMC outperforms HMC, which outperforms LMC in cases we study here. We are trading random walk of LMC with NF approximation error at a finite N, and we show that the tradeoff is in favor of DLMC, typically by one order of magnitude or more for sequential version, and many more orders for parallel version, in the limit of expensive likelihoods, where the cost of NF evaluation is small compared to the likelihood cost. We clearly show that the method converges to the correct posteriors with far fewer likelihood evaluations than state of the art baseline HMC, so our empirical results confirm the theory behind the method is sound. We perform ablation studies in appendix which show that preconditioning is useful and that we converge faster. The reviewer is questioning the motivation and arguing that the method is not useful or wrong, but the empirical evidence suggests otherwise. We will expand the discussion to clarify these points better.

---

> > ### Comment · Reviewer_yr5Z · 2022-08-07
> > **thank you for your rebuttal.**
> >
> > > There is no stochastic noise in our procedure (it is deterministic), so we are unclear why the reviewer talks about stochastic noise, simulating Gaussian noise, stabilizing noise procedure, writes down Wiener noise term dW_t etc. None of this is relevant for what we do.
> >
> > This is not a helpful response.  There is relevance in my review, time was taken out to carefully read it and present it and the nature of this response somewhat undermines it. Nonetheless I will take more time out to clarify this particular point.
> >
> > Yes the dynamics you work with are deterministic, however as you point out at the start of the paper there is a stochastic equivalent which has the same marginals from there you build the deterministic ODE with the FPK score as its vector field.  My point here is that the preconditioning you do (which has an auxiliary variables flavour) follows a determinsitic ODE which can be interepreted via an SDE in the exact same form you convert from langevin dynamics to an ODE with a score (this is just going backwards). The careful analysis I aimed in motivating to the reviewer with the hope of a fruitful discussion (rather than being discarded and told my review is not relevant to what you do) was to interpret the preconditioning in the langevin dynamics setting.
> >
> > To be more specific if I have a system of ODEs 1 for the preconditioner and one for the sampling dynamics this "deterministic" (initial condition is random() system admits an equivalent stochastic formulation in terms of langevin dynamics. Because the preconditioning felt poorly motivated in the initial draft of the paper I resorted to its stochastic interpretatoin (covnerted back to langevin dyanmics) to analyse the effect it had. I was indeed doubtful wheter p(z|y) was gaussian or not as the notation and setups in the text became somewhat contradictory, thank you for clarifying.
> >
> > > On preconditioning ...
> >
> > I understand well the benefits of preconditioning in different iterative schemes.  My point was that the particular nature of your preconditioning was not well explained in a self contained manner. From my experience in sampling and inference most preconditioniers that I understand well are either linear or carefully constructed (e.g. reimanian ULA by Girloami and Calderhead), preconditioning with flows on the otherhand is a bit more contemporary and should merit a self contained exposition rather than blame/point the reviewer to reference 14.
> >
> > > ... reviewer is questioning the motivation and arguing that the method is not useful or wrong,...
> >
> > Again this is another comment that is a bit defensive trying to discredit the review, rather than leveraging a discussion.  I want to highlight some issues with this response:
> >
> > 1. Even if empirical results are good, motivation is key. Empirical ablations whilst incredibly helpful should not be strong enough to make closing remarks on the lines of "my results are sota the reviewer should not criticise the preimse of the method". First of all since there are many other methods missing in the comparison, and the ablation is actually somewhat poor. ULA based samplers also admit preconditioniers a proper ablation of your  comparing to these could have helped close the argument. Additionally there are several methods (e.g. MCD [1], PIS [2,3] ) which combine langevin dynamics with neural samplers which are likely to outperform this approach (or be similar), so "good results" is really an insuficent argument, (also note the theory / gaurantees behind some of these methods is heavily and well motivated in their respective manuscripts).
> > 2. I want to re-iterate my main and central point. I never said the method was "wrong", I said that starting from langevin dyanmics which is something that we can construct a consistent approximation to (euler mayurama) with well undesrstood convergence rates, why would I transform this theoretical nice and well behaved SDE into an ODE that involves a score term which is very difficult to approximate, you are introducing a new source of bias / error into the system and loosing gaurantees. To give a very simple analogy imagine I have a linear model solvable via a pseudo-inverse (dxd) but instead I reinterpret it as a kernel method in a linear RKHS and solve by computing its gram-matrix (NxN) with N>>d , why would I trade for a representation that is much more taxing computationally and theoretically yields the same method ? In gen modelling settings (where we have available samples) it does make sense working with the deterministic system faster convergence as you motivated but also since the score admits incredibly sound (statiscally) and computational approximations.
> > 3. Finally to re-iterate I am also not saying the proposed preconditioning is wrong, but the way its presented it is confusing (and led me to think it was unclear/adhoc).

---

> > > ### Comment · Reviewer_yr5Z · 2022-08-07
> > > **part II**
> > >
> > > P.S some things have been better motivated in the rebutal, however unfortunately due to the above reasons I am not inclined to change my score for this current iteration of the manuscript.
> > >
> > > [1] Doucet, A., Grathwohl, W.S., Matthews, A.G.D.G. and Strathmann, H., 2022, March. Annealed Importance Sampling meets Score Matching. In ICLR Workshop on Deep Generative Models for Highly Structured Data.
> > >
> > > [2] Zhang, Q. and Chen, Y., 2021. Path Integral Sampler: a stochastic control approach for sampling. arXiv preprint arXiv:2111.15141.
> > >
> > > [3] Vargas, F., Ovsianas, A., Fernandes, D., Girolami, M., Lawrence, N. and Nüsken, N., 2021. Bayesian Learning via Neural Schr\" odinger-F\" ollmer Flows. arXiv preprint arXiv:2111.10510.

---

> > > > ### Author Response · Authors · 2022-08-09
> > > > **Thank you for further explanation of the comments**
> > > >
> > > > We thank the reviewer for the further explanation of the comments. We want to emphasize that in our original response we were not trying to discredit the review, and it was not our intention to be defensive: we simply failed to understand the specific points the reviewer was raising. For example, it was not clear to us that when discussing stochastic versions the reviewer was referring to a different method (that of classical Langevin MC), rather than to our DLMC method.
> > > >
> > > > As we currently understand it seems these are the main comments the reviewer has:
> > > >
> > > > 1) the reviewer is pointing out the papers which combine stochastic Langevin MC with neural samplers: we thank the reviewer for these references which we will add to the revised version. These papers are similar in spirit to reference 14 (Hoffman etal) which does neural Hamiltonian MC. It is unclear to us why the reviewer believes these references 1-3 will outperform our method: we believe that HMC outperforms LMC in most settings and that neural HMC outperforms neural LMC in most settings. Since we compare against neural HMC, and show DLMC outperforms it (due to the slow training of neural HMC with VI), we believe that DLMC is competitive against SOTA. We'd be happy to performs further ablation tests against these additional methods.
> > > >
> > > > The reason we do not emphasize more preconditioning or spend more time explaining it is that this is not a new idea, and that DLMC works also without it, i.e. it is not what distinguishes our paper from other work. The reviewer is suggesting we do spend more time and we will do so in the revisions. For example, Riemanian HMC of Girolami and Caldedhead is difficult to implement numerically because the Jacobian is position dependent. NFs have by construction simple Jacobians and thus can be viewed as a specific implementation of RHMC which can be simpler to implement.
> > > >
> > > > 2) We are arguing that replacing the stochastic term with a deterministic term has advantages, despite the fact that there is clearly a trade-off. The advantage is that DLMC method is deterministic, i.e. it has no noise: noise slows the convergence. This is well established in the context of HMC vs LMC: LMC can be viewed as a single leapfrog step of HMC, followed by stochastic  momentum resampling. Performing many leapfrog steps deterministically improves the convergence rate of HMC relative to a single step LMC. Recent work on generalized HMC makes a further step in this direction by making even the momentum resampling less stochastic (by partially preserving the direction, as pointed out by Horowitz, Neal etc). We go all the way by making it completely deterministic. The price we pay is that we have to estimate the current density with NFs. This is clearly a trade-off in that NFs are not perfect, but we show that the trade-off is beneficial in the examples where we tried it.
> > > >
> > > > We are arguing that our method is new, in that NF+deterministic Langevin have not been proposed before, and that it shows a lot of promise on realistic examples. We are not arguing that it will become the best sampler on the market, it is too early to say this, but it is also not clear that it cannot become SOTA. Furthermore, we believe the motivation for our method is clear: deterministic is good, random walk is bad for sampling, so how can we do it as deterministic as possible? Our proposed method is well defined and the trade-offs are clearly specified, and it feels too negative to question the whole premise of what we do by asking why would one want to ever do this. We think deterministic >> stochastic is a good motivation and the empirical results support this. We will emphasize this point further in revisions.

---

### Official Review · Reviewer_QrcG · 2022-07-11

**Rating:** 6
**Confidence:** 3
**Soundness:** 3 good
**Presentation:** 3 good
**Contribution:** 2 fair

**Summary:**

The work proposes a Bayesian inference procedure constructed using a deterministic Langevin equation. The resulting algorithm uses a number of particles initialized at random, their dynamics then following the deterministic equation toward its equilibrium distribution, while at regular time intervals a normalizing flow is fit to the collection of particles and used as an independence proposal within Metropolis-Hastings to propose replacing each particle with a new draw. The method is demonstrated on a number of toy examples and standard data sets.


**Questions:**

* Before the Metropolis-Hastings step, aren't the particles only approximately distributed according to the target distribution, given time discretization to get them there, and applying the Metropolis-Hastings kernel will not leave them properly distributed, either? This is not the case with HMC or MALA, say, where the time-discretized move is itself the proposal (over the current state of the Markov chain), and the Metropolis-Hastings rule accepts or rejects it. Here it seems there is a particle drawn from the time discretization, and a proposal drawn from the normalizing flow fit to the collection of those particles, and we accept or reject the swap. Perhaps the authors could better explain how this yields samples from the target distribution, or clarify the claims (e.g. first paragraph of conclusion claims "To make it asymptotically unbiased we add MH step...").

* Independence proposals rely on the proposal being a good approximation of the target distribution in order to have a reasonable acceptance rate, and so for the Markov chain to mix well. I can accept that a normalizing flow can achieve a good approximation with sufficient size, but I do not see any empirical results on this part of the algorithm, e.g. what is the average acceptance rate of this step in the examples?

* Bottom p.7 there is a comment that, "...we find there is a tradeoff between number of particles and number of iterations, such that starting with a larger number of particle does not always imply a higher overall computational cost." I can see how this might be the case, but can also seem the limits of it, as in e.g. the extreme case of reducing to one particle. A plot showing this tradeoff would be a good addition.

* As above, specific numerical results on the swap moves (e.g. acceptance rate) would be a good addition.

* As above, some clarity on the presentation of the parallel version.


**Limitations:**

No concerns.

**Strengths And Weaknesses:**

There are some interesting aspects to this work, such as the use of the deterministic Langevin equation, and the empirical results are quite good, especially the ablation studies in the supplementary material that separate out the various components that come together in the final algorithm. There are some gaps, however, that are stopping me from recommending acceptance.

While I think I understand the construction of the algorithm, it is not clear to me that this results in samples distributed according to the target distribution, even up to the approximations introduced (e.g. fitting the normalizing flow to the current stock of particles, time discretization). With that said, the empirical results suggest that the algorithm works as intended numerically; if there is a bias it may be small. I've added some questions for the authors on this topic below and hope they can help me to better understand their work.

It is unclear to me how the results for P-DLMC have been obtained (e.g. in Figure 3). The authors note that these are based "...on the assumption that we can evaluate the likelihood gradient in parallel on N CPU cores" (bottom p.7). Is there actually a parallel implementation here? In my opinion, unless there is a parallel implementation to provide empirical results, this ought to be an item for the discussion, not the experimental section. Otherwise, one is not accounting for the real-world challenges of parallel computing (fewer than N cores, possibly variable execution time between particles leading to load balancing issues, communication overhead in gathering all particles to update the normalizing flow, etc).

If it is a model rather than actual implementation, have the authors simply divided through by N? Or have they used a model where only some p portion of the sequential program can be parallelized, thus using a scaling factor such as (1 - p + p/N)? There is a comment about the assumption that evaluating the likelihood gradient takes about a minute, while the cost of the normalizing flow is in the seconds, but even that limits the maximum speedup to much less than N (say 60 seconds likelihood gradient parallelized down to essentially zero, 1 second for normalizing flow, that still limits speedup to factor of 61, i.e. 61 seconds down to 1 second, even when N >> 61). These are some of the challenges of scaling these sort of population/ensemble algorithms in practice (SMC has similar challenges).

---

> ### Author Response · Authors · 2022-08-01
> **MH and DL steps are independent and each converges to the target; MH is useful even when acceptance rate is low; P-DLMC implementation is embarrassingly parallel and gives acceleration roughly proportional to N (number of particles)**
>
> Q1: DL and MH steps are independent of each other, and each one separately converges to the true distribution under some technical conditions discussed below. They are unrelated to each other, and we can mix and match DL and MH steps, there is no need to view them in sequence.
>
> DL converges to the true distribution in the limit of large number of particles N and assuming NF is a universal approximator. See Moutsa et al. (ref 3) for related discussions in the context of KDE density. Under these conditions and for sufficiently small step size (related to the condition number, similar to the standard gradient based step size condition) the DL method converges, since the particles move only if the current NF density differs from the target, and the particles move in the direction of reducing the discrepancy. Once NF equals the target the gradient difference is zero (grad(U-V)=0) and particles no longer move. It is difficult to obtain any theoretical convergence rates
> for a finite N. However, it is generally true that at a fixed N NF is a better density estimator than KDE, which underlies Maoutsa et al. This is shown in appendix A, where we compare the performance of the two methods.
>
> DL only converges to the true distribution for simple unimodal distributions, and not for multiple disjoint components. This is because DL alone will get the particles into individual peaks, but since it is gradient based it  will not guarantee that their relative proportions are correct, and once they are trapped inside a peak DL cannot move them out. This is true for most samplers such as MH, HMC, LMC, which cannot handle multimodality in the absence of a separate algorithm such as annealing or tempering. With the addition of MH step we remove this limitation without having to resort to annealing or tempering, and we get it for free since NF is part of the algorithm already. MH step is thus needed in multi-modal settings for asymptotic convergence guarantees. MH step follows the same rules as other Metropolis-Hastings implementations and has the same convergence criteria: if we do MH step many times in a row we will converge. We found no need to do it in our examples, but can be done for additional convergence guarantees.
>
> Q2, 4: The reviewer is absolutely correct about the acceptance rate, which can be very low, especially in initial stages of the algorithm when
> the actual particle distribution is broader than the target. However, the role of MH step in these stages is simply to remove the worst particle performers. One can for example think of these as being stuck in a local maximum far from the global maximum, or more generally view these poor performers as particle optimization being dependent on initialization. MH removes these worst performers and replaces
> them with particles with a higher likelihood. We found that this really helps with the performance of DLMC, even though the MH acceptance rate may be low. We can add a figure showing how acceptance rate changes with number of iterations, but we stress that low acceptance rate of MH does not indicate slow convergence in DLMC, since it still helps the overall convergence rate of DLMC.
>
> Q3: We apologize for lack of clarity, all the examples assume N>2d, so the comment is meant in this context. We have not explored N<2d (e.g. N=1), since in this limit NF fails to converge to a meaningful density distribution. Ou remark was meant to convey that there is a broad range of values of N where the overall cost is approximately constant in the serial mode. In the parallel mode (P-DLMC), choosing a higher N is almost always beneficial, but we only tested this up to N=2000.
>
> Q5: P-DLMC: we have implemented embarrassingly parallel version of DLMC on NERSC. In our version we do not parallelize the individual likelihoods, but instead we run each likelihood on an individual core, with N cores, where N is number of particles. So for the example we show this is still 60 seconds for each likelihood, and the NF cost of 1 second is small compared to 60 seconds. Our implementation seems to differ from the reviewer's example, where the reviewer argues parallelization makes the cost of a single likelihood reduced to near zero: in our approach the speedup we obtain is close to N and is not limited to 60 in this example. Since all the cores are performing the same likelihood evaluation (with different parameter values) their evaluation time is nearly the same, so there are no load balancing issues. We included the overhead associated with communication and NF cost in our plot.

---

> > ### Comment · Reviewer_QrcG · 2022-08-08
> > **Thanks for the reponse**
> >
> > Thanks to the authors for their response. This addresses my concerns, especially Q1 and Q5, which were the more major in my opinion. I'll update my rating to reflect.

---

### Official Review · Reviewer_qZmF · 2022-07-14

**Rating:** 7
**Confidence:** 4
**Soundness:** 3 good
**Presentation:** 4 excellent
**Contribution:** 3 good

**Summary:**

The authors propose a mechanism to produce unbiased samples from expensive posteriors given that it is easy to sample from the prior and possible (although expensive) to evaluate the likelihood. The approach proposes concrete particle dynamics based on NFs that approach the target distribution and corrects them using Metropolis-Hastings.

**Questions:**

1. It would seem that a lot of the power of this approach is coming from the strong modeling ability of the NF. I am missing comparisons with simpler approaches that can leverage this power. For instance:

(a) What if we simply run variational inference to optimize the normalizing flow? One can easily compute the gradients with respect to the parameters. (For the SINF case, it might be possible to modify its main algorithm to take an unnormalized log-pdf instead of a set of samples - but the gradients for variational inference can always be approximated stochastically in a NF). Then one can find the closest q(z) within the NF domain and then sample from it and correct using MH. This decouples the number of particles used for learning and the ones used for sampling: More independent samples can be generated on demand after training. And does not require any particle dynamics.

(b) What if we only use SINF and MH? MH produces a new set of samples closer to the posterior, then we use those samples and SINF to update q(z) in closed form. Then we repeat. Again, we have eliminated the separate particle dynamics.

While (a) and (b) might be significantly worse than the proposal of the paper, it'd be reassuring to know that this is the case, and that all the heavy lifting is not being done by the NF being able to learn the energy function. Otherwise, the proposed dynamics don't seem justified enough.

2. The experiments in which the posteriors are calculated are pretty simple. No strong baselines for those models are used, just NUTS which of course is slow.

3. The tested models are _very_ low-dimensional. The high dimensional setting might have very poor performance, this is not known.

4. I expect multimodal likelihoods with broad priors to result in a lack of mixing - I doubt all modes will be discovered even in low dimensional cases, when they are spaced and sharp. Again this is not tested even in a 2-dimensional experiment with many gaussians that are well separated.


Minor comments:
- Parentheses are used for bibliography citations, should be square brackets, not to be confused with cited equations.
- In the MH update in Algorithm 1, time indices are missing.

[1] Black Box Variational Inference. Rajesh Ranganath et al.




**Limitations:**

Addressed in the previous section.




**Strengths And Weaknesses:**

Strengths:
- Technically correct
- Clear presentation
- Original dynamics
- Uses the gradient information of the target distribution
- Produces multiple uncorrelated samples

Weaknesses:
Experimental validation, in terms of:
- Complexity of the models sampled from
- Strength of the baselines
- Ablations

---

> ### Author Response · Authors · 2022-08-01
> **VI is inferior to DLMC; NUTS is the fastest baseline; 50-100 dimensional non-Gaussian posteriors are common and not easy; global optimization is an unsolved problem**
>
> Q 1: VI with NF has been tried in many previous papers, and the overall conclusion is that it often does not converge to the true target, even when the approximating NF function is sufficiently powerful. One example is Hoffman et al paper (ref 14), where they pretrain using VI with an enormous training time (they use 20 million likelihood calls). They still need to follow up with the preconditioning method, using HMC sampling in NF latent space. Another example is Modi, Li, Blei https://arxiv.org/pdf/2206.15433.pdf, where they point out the mode seeking nature of reverse KL divergence underlying VI leads to mode collapse (i.e. too marrow posterior) even though NF is sufficiently powerful.
>
> With regard to Q1(b) SINF+MH, we perform this ablation study in the supplementary material, fig. 9. We show DL step is required for a fast convergence. We believe these examples show that particle dynamics based NF methods are superior to pure VI+NF.
>
> Q2: NUTS is a state of the art baseline that is widely used (e.g. in STAN, PyMC3 etc.). We have followed the recent Hoffman et al (ref 14), which only compare to NUTS. We are not aware of any other sampler that consistently outperforms NUTS in terms of evaluation time.
>
> Q3: Most of our examples are 50-100 dimensional, with very non-Gaussian posteriors. They are common and they are not easy: most samplers struggle on these examples and fail, or take a very long time. We followed recent literature like Hoffman et al (ref 14) in terms of examples and dimensionality. We have evidence that our method works on higher dimensional problems with simple unimodal posteriors, and we can add an example if the reviewer specifies the specifics (target and dimensionality).
>
> Q4: When it comes to multimodal problems there is no method that can discover the true global maximum: global non-convex optimization is an unsolved problem. Moreover, the no free lunch theorem of global optimization informs us that there is no single method that can be fast and correct on all examples: one either chooses to do expensive exploration, wasting computational time on simpler problems, or one chooses to do fast exploitation, risking to miss other peaks on hard problems. Given that one can come up with an arbitrarily hard problem we are unsure what are we supposed to show, but we'd be happy to do another example if the reviewer specifies it. What we wanted to emphasize here is that our method can handle multimodal posteriors with no additional modifications to the algorithm, in contrast to standard samplers (MH, HMC, LMC), which cannot handle multimodality without additional methods such as annealing/tempering. To demonstrate this a simple double peak example suffices.
>
> We will fix the parentheses and clarify better the time index notation.

---

### Meta-Review · Area_Chair_Yji4 · 2022-08-26

**Recommendation:** Accept
**Confidence:** Less certain

**Metareview:**

This paper proses an inference method that combines gradient ascent and normalizing flows. The idea is that one could, in principle, simulate the deterministic Fokker-Planck equation, but this would require access to the density of the evolving approximating density, which is intractable. Thus, the paper proposes to maintain a set of particles and update a normalizing flow to approximate this density. The resulting procedure is deterministic, with an accuracy that depends on the number of particles and the power of the normalizing flow.

Reviewers agreed this was an interesting approach and the experimental results are promising (albeit fairly low-dimensional). However, there were a few apparent weaknesses: Firstly, there was a lack of clarity about the theoretical guarantees. The authors state that this is all clear in the manuscript, but readers would undoubtedly benefit from a much more centralized/explicit description of what approximations are involved, and under what guarantees the method is claimed to work, which could be put in a single place. In addition, in trying to understand exactly what was done in the experiments, it is difficult to understand several of the details. Algorithm 1 is very helpful in this regard, but it would be beneficial to have a self-contained elaboration of all of the points (perhaps even in an appendix). Finally, the experimental results are all relatively low-dimensional. Often particle methods do not scale gracefully to higher dimensions. (I do not see this as a huge flaw because the method could be useful even if it does not scale, but reader would benefit from evidence on this point either way.)

In the end, however, most of the above issues are issues of clarity and I am willing to trust the authors to fix these before final submission. The paper appears to present a novel idea and the community would benefit from seeing it and discussing it.

**Award:**

No

---

### Decision · Program_Chairs · 2022-09-14

Accept